# The Association between Race and Survival among Pediatric Patients with Neuroblastoma in the US between 1973 and 2015

**DOI:** 10.3390/ijerph17145119

**Published:** 2020-07-15

**Authors:** Farouk S. Farouk, Omar A. Viqar, Zaid Sheikh, Grettel Castro, Noël C. Barengo

**Affiliations:** 1Department of Translational Medicine, Herbert Wertheim College of Medicine, Florida International University, Miami, FL 33199, USA; ffaro001@med.fiu.edu (F.S.F.); oviqa001@med.fiu.edu (O.A.V.); zshei001@med.fiu.edu (Z.S.); gcastro@fiu.edu (G.C.); 2Department of Public Health, Faculty of Medicine, University of Helsinki, 00014 Helsinki, Finland; 3Department of Public Health and Epidemiology, Riga Stradins University, LV-1007 Riga, Latvia

**Keywords:** neuroblastoma, neoplasms, child, ethnic groups, epidemiology, mortality, population cohort

## Abstract

*Background*: Conclusive information regarding the influence of race on survival among neuroblastoma patients is limited. Our objective is to investigate the association between race and cause-specific survival in pediatric patients diagnosed with neuroblastoma in the US between 1973 and 2015. *Methods*: This was a retrospective cohort study using the Surveillance, Epidemiology, and End Result (SEER) database. Patients aged 17 and younger of black, white, or Asian Pacific Islander (API) race diagnosed with neuroblastoma from 1973–2015 were included (n = 2,119). The outcome variable was time from diagnosis to death. Covariates included age, gender, ethnicity, stage, tumor site, and year of diagnosis. Cox proportional hazard models were used to calculate hazard ratios and 95% confidence intervals. *Results*: There were no statistically significant differences in the hazard of survival for blacks (HR 0.93; 95% confidence interval (CI) 0.74–1.16) or API (HR 1.02; 95% CI 0.76–1.37) compared with whites. However, patients diagnosed between 2000–2004 (HR 0.46; 95% CI 0.36–0.59) and 2005–2015 (HR 0.33; 95% CI 0.26–0.41) had decreased hazards of death when compared to patients treated during 1973 to 1999. *Conclusions*: No association between race and survival time was found. However, survival improved among all patients treated during 2000–2004 and 2005–2015 compared with those treated before the year 2000, leading to a narrowing of the racial disparity based on survival.

## 1. Introduction

Among the pediatric population, cancer is the leading cause of disease-related death [1]. Neuroblastoma is the most common childhood cancer diagnosed during the first year of life, and the third most commonly diagnosed childhood cancer overall. Overall survival rates for children with neuroblastoma have seen an upward trend [2]. Yet, the prognosis for neuroblastoma patients depends on various factors such as age of the child when diagnosed, and the stage, the molecular, and cytological characteristics of the tumor. Moroz et al. revealed that even though adverse influence of increasing age-at-diagnosis has declined, it remains a powerful indicator of unfavorable prognosis [3].

Racial disparities in cancer survival have been extensively researched for hematological malignancies, but less so for solid tumors, including neuroblastoma [4]. Whites have a higher incidence of neuroblastoma than blacks and other races [5,6]. Possible reasons for such disparities in incidence of neuroblastoma include differences in socioeconomic status, delays in diagnosis because of inadequate access to care, lack of cancer knowledge on the part of patients, risky health behaviors, and differing disease biology [7].

Despite racial differences reported in the incidence of neuroblastoma, there is not much research on how race influences survival among children with neuroblastoma [8,9,10,11,12,13,14]. Results in regard differences in survival between white and black patients have been inconsistent [8,11,12,13,14]. Whereas some of the previous studies using SEER data revealed a reduced survival in Back patients [10,11], others did not coincide [9,12,14]. A study of 3539 children diagnosed with neuroblastoma and participating in the Children’s Oncology Group neuroblastoma biology protocol between 2001 and 2009 showed that compared to whites, blacks had a 1.4-fold risk of death due to neuroblastoma, while Asians and Native Americans had a 1.6 and a 3-fold increased risk, respectively. While blacks and whites were the most commonly included in studies, studies that also assess Native Americans and Asians [8,9], Hispanics [8,9,12], American Indians and Native Pacific Islanders [9,12] survival in neuroblastoma patients are scarce [8,9,10,12].

The objective of the present study was to determine if there is an association between race and 5-year cause-specific survival in pediatric patients diagnosed with neuroblastoma in the United States between 1973 and 2015.

## 2. Materials and Methods

### 2.1. Study Design and Population

A secondary analysis of data from the National Cancer Institute’s Surveillance, Epidemiology, and End Result (SEER) Database was performed. Briefly, SEER is a surveillance system that collects and publishes cancer incidence and survival data from population-based cancer registries throughout the US. Registries comprise 34% of the US population and originate from 19 geographical areas [15].

For this study, children aged 17 years or younger diagnosed with a first-time diagnosis of neuroblastoma (ICD-O-3 9500/3) from 1973 to 2015 were included. After exclusion of participants missing information on survival, and participants for which diagnosis was done at autopsy, and duplicate information, the final sample size was 2119.

### 2.2. Main Variables

The primary outcome was cause-specific survival, defined as the time in months from diagnosis to death due to neuroblastoma, which was recorded up to 60 months. This was calculated using the SEER cause-specific death classification (dead, alive/dead of other cause) and survival time (in months). The independent variable of interest was race categorized as white, black, or Asian Pacific Islanders (API). Other variables assessed included sex (male/female), ethnicity (Hispanic, Non-Hispanic), participant’s age (categorized into <1 year, 1 to 4 years, and 5 to 17 years), year of diagnosis, cancer stage (localized, regional, distant, unstaged/unknown), and primary neuroblastoma site (adrenal or non-adrenal sites) according to ICD-0-3 codes available at the SEER database. Adrenal site was considered if codes ranged from C740–C749. Non-adrenal sites category included neuroblastoma of the mediastinum (codes C381 to C383), autonomic nervous system (codes C470-C479), retroperitoneum (code C480), connective tissue (codes C490 to C499), and central nervous system (codes C700 to C729). These age group categorization was based on survival prognosis previously described [3]. Pinto et al. showed that survival rates were determined to be significantly different according to diagnostic era with better outcome observed for patients diagnosed after 2000, when consolidation with HDT and stem-cell rescue was routinely included in the treatment plan for high-risk patients [16]. Only 6% (64 of 1015) of the patients diagnosed between 2000 and 2004 and 30% (445 of 1484) of those diagnosed between 2005 and 2010 received immunotherapy and cytokines plus isotretinoin after consolidation. Postconsolidation treatment with immunotherapy and cytokines plus isotretinoin is now considered part of standard-of-care treatment. Therefore, the categories of year of diagnosis, were the intervals of 1979 to 1999, 2000 to 2004, and 2005 to 2015 as to coincide with changes in treatment protocol for neuroblastoma implemented in 2000 and 2005 that could impact patient’s outcomes [16].

### 2.3. Analysis

The data were exported from SEER and analyzed with Stata SE version 14 (StataCorp. 2015. *Stata Statistical Software: Release 14*. College Station, Texas, TX: StataCorp LP, USA). All data accessed from SEER were de-identified (fully anonymized) according to the Health Insurance Portability and Accountability Act. Stata software version 15 was used for all analyses [17]. We performed descriptive statistics to assess the frequency distributions of the variables and patterns of missing data. Bivariate analyses were conducted to compare the distribution of potential confounders by race. Chi squared tests were used to assess statistical significance. Kaplan Meier curves were generated by selected independent variables (by race and year of diagnosis) and log-rank tests were performed to assess for statistical differences between the survival curves across the different independent variables. Cox proportional hazards regression models were used to calculate the unadjusted and adjusted hazard ratios with their corresponding 95% confidence intervals. Statistical significance was considered for *p*-value of less than or equal to 0.05 (two tailed tests).

### 2.4. Ethical Considerations

This study was reviewed by the Florida International University (FIU) Office of Research Integrity and determined to be Not Human Subject Research. As a result, the study did not require submission to and approval of the FIU Institutional Review Board (IRB). The privacy of all participants was maintained as all information utilized was secondary de-identified data obtained from SEER.

## 3. Results

Table 1 describes the demographics and clinical characteristics for patients with neuroblastoma in the SEER registry from 1973–2015. When compared with whites and API patients, a significantly higher proportion of black patients were Non-Hispanic (89% vs. 95.8% vs. 97.5%) and were diagnosed with neuroblastoma at 1–4 years old (55.8%). A lower proportion of black patients were <1 year of age at presentation compared with white and API. Also, a higher proportion of API patients had adrenal disease compared with white and black patients. A higher proportion of API patients, when compared to whites and blacks, had a distant stage at diagnosis (44.9% vs. 40.6% vs. 37.2%) and were diagnosed between 2005 and 2015 (40.6% vs. 27.3% vs. 34.7%). More API had adrenal neuroblastomas than whites and blacks, respectively (53.9% vs. 40.6% vs. 40%).

Figure 1 displays the Kaplan Meier adjusted survival curves according to race from 1973–2015. Survival was statistically significantly different among the year of diagnosis. Analysis of the data did not show statistically significant differences in survival between the three racial groups (log rank test *p*-value 0.11).

Table 2 shows the unadjusted and adjusted hazard ratios for 5-year cause-specific survival in the study sample. The final adjusted model included age, gender, ethnicity, stage, site, and year of diagnosis. These covariates were identified as being clinically relevant and were sued in previous studies as well [8,9,10,11,12,13,14]. In addition, they were unequally distributed according to the race (Table 1) and the survival. After adjusting for these covariates, there were no statistically significant differences in the hazard of survival for blacks (hazard ratio (HR) 0.93; 95% confidence interval (CI) 0.74–1.16) or API (HR 1.02; 95% CI 0.76–1.37) when compared with whites. Results were also not significant for female (HR 0.89; 95% CI 0.76–1.04), Hispanic ethnicity (HR 1.00; 95% CI 0.75–1.34), and adrenal primary tumor site (HR 1.16; 95% CI 0.99–1.36). Patients 1–4 and 5–17 years-old had, respectively, a 5.92-fold (95% CI 4.64–7.56) and 7.29-fold (95% CI 5.51–9.66) greater hazard of death when compared with those <1 years-old. Lastly, distant stage had the higher death hazard compared to the localized stage (HR 9.03; CI 4.94–16.49). Lastly, patients diagnosed between 2000–2004 (HR 0.46; 95% CI 0.36–0.59) and 2005–2015 (HR 0.33; 95% CI 0.26–0.41) had significantly lower hazards of death when compared with patients diagnosed from 1973 to 1999.

Figure 2 displays the adjusted hazard ratios of race, ethnicity, sex, type of cancer and year of diagnosis for 5-year cause-specific survival among patients in the SEER database diagnosed with neuroblastoma between 1973 and 2015. Patients diagnosed between 2000–2004 and 2006–2015 had increased survival compared with patients diagnosed during 1973–1999.

## 4. Discussion

The results showed no association between race and survival among pediatrics patients diagnosed with neuroblastoma, survival improved among all patients diagnosed after the year 2000.

Our results update and expand the results reported in previous studies using SEER [10,11,18]. A study analyzing survival between black and white patients with childhood cancer from 1992 to 2000 and from 2001 to 2007 revealed that despite significantly improved treatment outcomes, black patients had significantly poorer rates of survival in both study periods compared with white patients [11]. Moreover, Kehm et al. found that black neuroblastoma patients exhibited 38% higher risk of mortality compared with white patients after adjusting for age, sex, and stage of diagnosis [18]. However, the emphasis of their paper was more on the role of socioeconomic status on outcomes and highlighted the mediating fore of socioeconomic status when looking at race. Finally, research comparing the survival rates by race among children with common extracranial solid tumors between 1985 and 2005 reported that black children had a 31% higher risk of death compared with white children [10]. Studies that used other databases than SEER reported inconsistent results with regard to an association between race and survival in children with neuroblastoma [8,12,13,14]. In the early 1970s, DiNicola et al. compared the survival rates of 30 white and 15 black children with neuroblastoma treated at the Children’s Hospital National Medical Center in Washington DC [14]. They did not reveal a statistically significant difference between the white and black children regarding median duration of survival or percentage of long-term survivors [14]. Henderson et al. showed a significantly increased risk of death from neuroblastoma in blacks compared to whites, before adjustment for risk groups [8]. After adjustment, overall survival did not differ between white and black patients. However, among patients who remained event-free for 2 years or longer, survival was worse in blacks compared with whites (HR 1.5, 95% CI 1.0–2.3) [8]. Also, a US cohort study including 2709 patients revealed that African genomic ancestry was associated with a 40% lower event-free survival demonstrating that common genetic variation influences neuroblastoma phenotype and contributes to the ethnic disparities in survival observed [13]. Finally, a recent study in patients diagnosed with neuroblastoma between 1986 and 2012 using data from the Therapeutically Applicable Research to Generate Effective Treatments initiative did not find any statistically significant differences in survival between black and white patients [12].

Several explanations for our observations can be elucidated. Our data from the SEER database included more recent patient cases, particularly from 2010 onwards. Previous literature had access to data as recent as 2009. This, combined with continued changes in treatment protocols for neuroblastoma in 2000 and 2005, may have led to a narrowing of the racial disparity based on survival [16]. This hypothesis is supported by our findings of significantly increased survival among all patients from the year 2000 on. The improved survival could have been the result of changes in neuroblastoma treatment protocols. Before the year 2000, the standard of care included high-dose chemotherapy (HDT) with stem-cell transplantation. In the latter time period, immunotherapy and cytokines plus isotretinoin were added to the treatment regimen [16]. Thus, these changes in treatment regimens could have led to better outcomes. Additionally, improved access to healthcare among patients of all races may have also played a role in the difference of this study with those using data from earlier years.

Our analyses also showed that an adrenal source of neuroblastoma was not significantly associated with survival when compared to a non-adrenal source, which was not expected. Previous literature has shown that an adrenal site is associated with worse survival than a non-adrenal site, but our results were not consistent with that [19]. A study of children younger than age 21 years diagnosed with NB or ganglioneuroblastoma between 1990 and 2002 and with known primary site were identified from the International Neuroblastoma Risk Group database has shown that patients with primary adrenal tumors have a statistically significant worse event free survival and overall survival compared to non-adrenal sites (HR 1.17; CI 1.05–1.29) even after adjustment for age, *MYCN* proto-oncogene status, and stage [20]. Possible reasons for the varying results could have been due to the different categorization and use of patient characteristics, neuroblastoma sites, time periods, and the use of different databases. For example, Vo et al. used patients up to the age of 21, included patients with ganglioneuroblastoma, investigated data from 1990 to 2002, and used patients enrolled in the International Society of Pediatric Oncology Europe Neuroblastoma group, Localized Neuroblastoma European Study, a North American Children’s Oncology group study, and trials from other countries [20]. In our study, we categorized our neuroblastoma sites as follows: adrenal, mediastinum, ANS, retroperitoneum, connective tissue, and CNS, before simplifying it to adrenal vs. non-adrenal neuroblastoma. Other studies chose to organize by gross anatomic classification. For example, Vo et al. used adrenal, abdominal/retroperitoneal, neck, thoracic, pelvic, and other sites [20]. Furthermore, increased stage and distant site are often useful predictors of worsening survival among cancer patients.

Some limitations should be considered. The SEER database only records information from the following geographic locations: Connecticut, New Mexico, Iowa, California, Kentucky, New York, Utah, Georgia, Hawaii, Massachusetts, Wisconsin, Detroit, and Louisiana [10]. There is also incompleteness in patient information with respect to treatment course, family income, or family history which limits the adjustment of these variable [17]. In addition, misclassification of race is possible, and it could lead to underestimation of differences in survival between blacks and whites. A similar limitation was reported with regards to the over-classification of Hispanic patients by the SEER database for patients between 1975 and 1999, possibly underestimating differences in survival among Hispanic and Non-Hispanic pediatric patients with cancer [8,21]. Moreover, insurance status information was not available prior to 2007 and previous studies have shown that black children with acute lymphocytic leukemia, who had worse outcomes in survival were less likely have private insurance and more likely to have public insurance [4]. Furthermore, we did not have access to information related to variables such as socioeconomic status and education. Social determinants may be the underlying factors in racial disparities of survival in neuroblastoma patients, and future research may study these factors in more detail. We are also aware that according to international consensus an age cut-off of 1-year-old is not standard and that 1.5 years should be considered [22]. However, age is collected in SEER as a discrete variable measured in full years. Thus, it was not possible to create age a cut-off value of 1.5 years-of-age. In order to find out whether this altered our results, we run additional analysis excluding children below the age of 1 (*n* = 877). The HRs and 95% CI remained statistically non-significant. Lastly, there are some latent biological factors beyond those available in SEER that are known to affect survival in neuroblastoma patients and may be potentially confounded with race, such as *MYCN*, ploidy, and tumor histology, among others. Randomized clinical trials are better study designs to control for confounders than observational studies. However, because of the nature of our exposure a randomized clinical trial is not applicable for our research question. Thus, best scientific evidence needs to be derived from future cohort studies that may control for most of the above-mentioned confounders we did not have available.

## 5. Conclusions

In conclusion, this study found no evidence of an association between race and survival among pediatric patients diagnosed with neuroblastoma. Future studies should have a closer look at what other factors, socioeconomic or otherwise, may influence survival among this patient population. In our analysis, insurance was used as a surrogate for socioeconomic status when assessing its influence on the relationship between race and survival. The fact that the entire pediatric population had at least some form of insurance and that there was no statistical significance in mortality based on the type of insurance serves as a compelling reason to employ universal healthcare. In addition, research that can address and fix the limitations of ours and previous studies in order to compare results and make a more definitive claim about the association between race and survival should still be pursued. Finally, the change in standard of treatment after 2000 showed a statistically significant decrease in hazard ratio. Thus, survival improved among all patients treated during 2000–2004 and 2005–2015 compared with those treated before the year 2000, leading to a narrowing of the racial disparity based on survival. Improvements in treatment specific protocols should be pursued for all patients to benefit overall survival outcomes.

## Figures and Tables

**Figure 1 ijerph-17-05119-f001:**
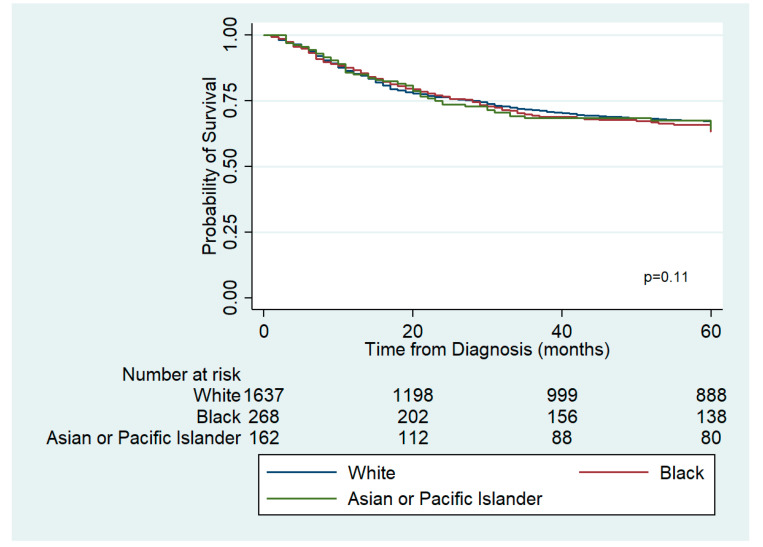
Adjusted survival curves according to race for patients in the SEER database diagnosed with neuroblastoma, 1973–2015.

**Figure 2 ijerph-17-05119-f002:**
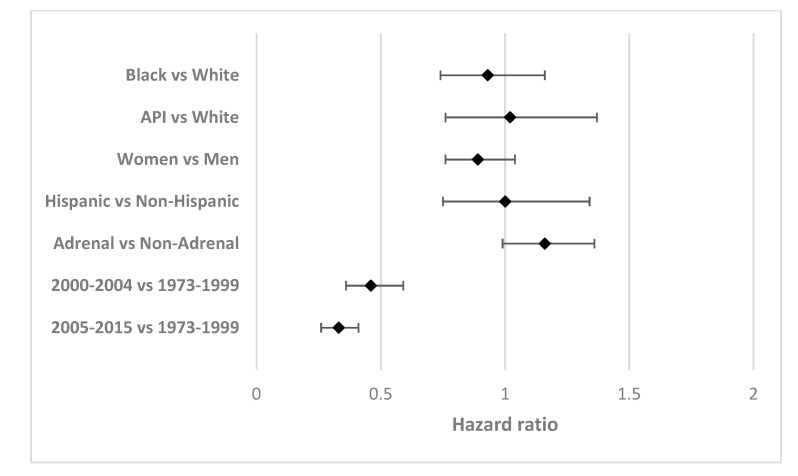
Forest plot of the adjusted hazard ratios of race, ethnicity, sex, type of cancer, and year of diagnosis for 5-year cause-specific survival among patients in the SEER database diagnosed with neuroblastoma between 1973 and 2015.

**Table 1 ijerph-17-05119-t001:** Demographic and clinical characteristics, by race, of pediatric patients in the SEER registry diagnosed with neuroblastoma, 1973–2015.

Characteristics	White (*n* = 1680)		Black (*n* = 274)		API ^a^ (*n* = 165)		*p*-Value
	% (*n*)		% (*n*)		% (*n*)		(*n*)
Age (years)							0.001
<1	43.1	−724	29.6	−81	43.6	−72	
1–4	46.5	−781	55.8	−153	45.5	−75	
5–17	10.4	−175	14.6	−40	10.9	−18	
Gender							0.661
Male	54.6	−917	52.6	−144	57	−94	
Female	45.4	−763	47.5	−130	43	−71	
Ethnicity							<0.001
Hispanic	11.3	−190	2.55	−7	4.2	−7	
Non-Hispanic	89	−1490	97.5	−267	95.8	−158	
Stage							0.415
Localized	10.7	−180	9.5	−26	10.3	−17	
Regional	18.3	−308	17.2	−47	14	−23	
Distant	40.6	−682	37.2	−102	44.9	−74	
Unstaged/Unknown	30.4	−510	36	−99	31	−51	
Site							0.004
Adrenal	40.6	−682	40.5	−111	53.9	−89	
Non-Adrenal	59.4	−998	59.5	−163	46.1	−76	
Year							0.001
1973–1999	59.5	−999	52.6	−144	46.7	−77	
2000–2004	13.3	−223	12.8	−35	12.7	−21	
2005–2015	27.3	−458	34.7	−95	40.6	−67	

^a^ Asian Pacific Islander.

**Table 2 ijerph-17-05119-t002:** Unadjusted and adjusted hazard ratios for 5-year cause-specific survival among patients in the SEER database diagnosed with neuroblastoma between 1973 and 2015.

Characteristics	Unadjusted	Adjusted
	HR ^a^ (95% CI ^b^)	HR (95% CI)
Race		
White	Ref.^c^	Ref.
Black	1.04 (0.83–1.30)	0.93 (0.74–1.16)
API ^d^	1.02 (0.76–1.35)	1.02 (0.76–1.37)
Age		
<1 years	Ref.	Ref.
1–4 years	6.65 (5.22–8.47)	5.92 (4.64–7.56)
5–17 years	8.57 (6.49–11.31)	7.29 (5.51–9.66)
Sex		
Male	Ref.	Ref.
Female	0.86 (0.74–1.00)	0.89 (0.76–1.04)
Ethnicity		
Non-Hispanic	Ref.	Ref.
Hispanic	0.77 (0.58–1.02)	1.00 (0.75–1.34)
Stage		
Localized	Ref.	Ref.
Regional	3.63 (1.91–6.88)	3.06 (1.61–5.80)
Distant	13.45 (7.39–24.50)	9.03 (4.94–16.49)
Unstaged/Unknown	7.78 (4.24–14.29)	7.32 (3.97–13.51)
Site		
Non-Adrenal	Ref.	Ref.
Adrenal	1.37 (1.18–1.59)	1.16 (0.99–1.36)
Year		
1973–1999	Ref.	Ref.
2000–2004	0.51 (0.40–0.65)	0.46 (0.36–0.59)
2005–2015	0.39 (0.31–0.48)	0.33 (0.26–0.41)

^a^ Hazard Ratio; ^b^ Confidence Interval; ^c^ Reference Group; ^d^ Asian Pacific Islander.

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
