# Peer review of "The Association between Race and Survival among Pediatric Patients with Neuroblastoma in the US between 1973 and 2015"

_ijerph, 2020, doi:10.3390/ijerph17145119_

Round 1
Reviewer 1 Report
The authors have fully addressed my comments and revised the manuscript according to my suggestions. The data is in an acceptable format now and the conclusions/interpretations are clearly presented. The results can now be compared to previous analyses, which has been discussed appropriately in the revised version of the text.
Author Response
Thank you.
Reviewer 2 Report
Overview
The results are misleading due to the censoring of neuroblastoma recurrences. This reviewer is uncertain whether these results accurately reflect outcome disparities by race, and is extremely skeptical of the authors’ conclusions.
Major
What is the justification/reasoning for censoring patients at the time of recurrence of neuroblastoma (NB)? This is a very non-standard way to perform an analysis of overall survival (OS). In fact, death due to NB is extremely likely to occur after recurrence. Also, there is a great loss of information with this approach, as the analysis is powered by “events” (i.e., deaths). Unless a very good reason for censoring NB recurrences can be provided, this is viewed as a fatal flaw in the analytic methods. The results are likely to be different if the censoring for NB recurrence is removed. Perhaps the disparities for race are magnified after the time of recurrence? Furthermore, the resulting OS curves are very misleading (too high), as many more patients died from NB than is reflected by the curves, due to the censoring at the time of NB recurrence.
An age cut-off of 1 year is non-standard; 1.5 years should be considered. See London et al (JCO 2005). The reference to Reis et al [3] is outdated. In 2005, an age cut-off of 1.5 years was adopted by international consensus.
Minor
Line 46 – the authors may also wish to consider Moroz et al (EJC 2011) as a more comprehensive analysis of the prognostic strength of age in neuroblastoma over time.
References
London WB, Castleberry RP, Matthay KK, Look AT, Seeger B, Shimada H, Thorner P, Brodeur G, Maris JM, Reynolds CP, Cohn SL. 2005. Evidence for an age cut-off greater than 365 days for neuroblastoma risk group stratification in the Children’s Oncology Group. Journal of Clinical Oncology 2005 Sep 20;23(27):6459-65. (highlighted cover story) PMID: 1611653
Moroz V, Machin D, Faldum A, Hero B, Iehara T, Mosseri V, Ladenstein R, De Bernardi B, Rubie H, Berthold F, Matthay KK, Monclair T, Ambros PF, Pearson ADJ, Cohn SL, London WB. Changes over three decades in outcome and the prognostic influence of age-at-diagnosis in young patients with neuroblastoma: A report from the International Neuroblastoma Risk Group Project.
European Journal of Cancer. 2011 Mar; 47(4):561-71. Epub 2010 Nov 26. PMID: 21112770
Author Response
Thank you very much for all your valuable comments and guidance to improve our manuscript. We have carefully studied your comments and revised our work according to your suggestions. Your feedback helps us to become better researchers and are highly appreciated.
Comment#1: What is the justification/reasoning for censoring patients at the time of recurrence of neuroblastoma (NB)? This is a very non-standard way to perform an analysis of overall survival (OS). In fact, death due to NB is extremely likely to occur after recurrence. Also, there is a great loss of information with this approach, as the analysis is powered by “events” (i.e., deaths). Unless a very good reason for censoring NB recurrences can be provided, this is viewed as a fatal flaw in the analytic methods. The results are likely to be different if the censoring for NB recurrence is removed. Perhaps the disparities for race are magnified after the time of recurrence? Furthermore, the resulting OS curves are very misleading (too high), as many more patients died from NB than is reflected by the curves, due to the censoring at the time of NB recurrence.
Response#1: Thank you for your comment. Apparently, there was a misunderstanding due to how we described this in the manuscript. We meant that we excluded duplicate information but did not censor these patients.
We have now revised the sentence in the materials and methods section of the manuscript to describe better the inclusion and exclusion criteria. We hope this clarified your concern.
The sentence states now as follows:
“For this study, children aged 17 years or younger diagnosed with a first-time diagnosis of neuroblastoma (ICD-O-3 9500/3) from 1973 to 2015 were included. After exclusion of participants missing information on survival, and participants for which diagnosis was done at autopsy, and finally those with duplicate information (patients who were diagnosed with a recurrence of neuroblastoma), the final sample size was 2,119.”
As we did not censor patients at the time of recurrence of neuroblastoma, the curves are not misleading.
Comment#2: An age cut-off of 1 year is non-standard; 1.5 years should be considered. See London et al (JCO 2005). The reference to Reis et al [3] is outdated. In 2005, an age cut-off of 1.5 years was adopted by international consensus.
Response#2: Thank you for this valid comment. Unfortunately, the data for age in SEER was collected as a discrete variable measure in whole years. Thus, it is not possible to select patients 1.5 years or older. However, we did additional Cox regression analysis excluding patients younger than 1- year-old (n=877) to calculate adjusted hazard ratios (HR) and their corresponding 95% confidence interval (CI). The HR and 95% CI did not change as can see in the table below:
--------------------------------------------------------------------------------------------
_t | Haz. Ratio Std. Err. z P>|z| [95% Conf. Interval]
---------------------------+----------------------------------------------------------------
V96 |
Black | .9168745 .1093963 -0.73 0.467 .725687 1.158432
Asian or Pacific Islander | 1.050121 .1674503 0.31 0.759 .7682603 1.435391
In order to address this difference in our manuscript compared with the international consensus, we have added the following explanation to the limitations section of the discussion (we also added the recommended reference):
“We are aware that according to international consensus an age cut-off of 1-year-old is not standard and that 1.5 years should be considered [22]. However, age is collected in SEER as a discrete variable measured in full years. Thus, it was not possible to create age a cut-off value of 1.5 years-of-age. In order to find out whether this altered our results, we run additional analysis excluding children below the age of 1 (n=877). The HRs and 95% CI remained statistically non-significant.”
Reference added: 22. London, W.B.; Castleberry, R.P.; Matthay, K.K.; Look, A.T.; Seeger, B.; Shimada, H.; Thorner, P.; Brodeur, G.; Maris, J.M.; Reynolds, C.P.; Cohn, S.L. Evidence for an age cut-off greater than 365 days for neuroblastoma risk group stratification in the Children’s Oncology Group. Journal of Clinical Oncology 2005, 23(27), 6459-65.
Comment#3: Line 46 – the authors may also wish to consider Moroz et al (EJC 2011) as a more comprehensive analysis of the prognostic strength of age in neuroblastoma over time.
Response#3: We have cited the work by Moroz et al in line 46. Following sentence has been added:
Moroz et al revealed that even though adverse influence of increasing age-at-diagnosis has declined, it remains a powerful indicator of unfavorable prognosis [3].
Reference added:
Moroz, V.; Machin, D.; Faldum, A.; Hero, B.; Iehara, T.; Mosseri, V.; Ladenstein, R.; De Bernardi, B, Rubie, H.; Berthold, F.; Matthay, K.K.; Monclair, T.; Ambros, P.F.; Pearson, A.D.J.; Cohn, S.L.; London, W.B. Changes over three decades in outcome and the prognostic influence of age-at-diagnosis in young patients with neuroblastoma: A report from the International Neuroblastoma Risk Group Project. European Journal of Cancer 2011, 47(4), 561-71.
Reviewer 3 Report
I would like to congratulate the authors on their manuscript on the association between race and survival among pediatric patients with neuroblastoma in the US based on SEER data. The paper is well written, complete, and logical. The methods are also appropriate. Having conclusive information on the role of race as a reflection of biology or socioeconomics remains a challenge and this manuscript contributes to elucidating possible answers.
General suggestions to improve the manuscript are as follows:
- Insurance status – The element of insurance status is the only truly inconsistent element of the Methods/Results/Discussion. In the current version, insurance status appears as a finding in the discussion without provision of enough detail in the methods and results sections. I found it hard to determine if it should stay or be removed from the manuscript all together because the information provided is too limited and vague. If the authors want to retain this element, it should be mentioned as a variable in section 2.2 in greater detail. It seems that data is only available for 2007-2015, that should be specified there as well. Mentioning sub-analysis in section 2.3 needs a bit more detail, was this pursued only as univariate analysis? Was there a multivariate analysis of the 2007-2015 subset? Distribution should be shown in table 1 and results should be mentioned in the results section.
- Methods – SEER*Stat – Where the analyses done using SEER*STAT software or exported and analyzed in an external software. The software and version should be specified.
- Results – The sentence that spans lines 126-128 on stage could be removed as those differences are not significant. I would highlight that a lower proportion of black patients were <1 year of age at presentation compared to White and API and that a higher proportion of API patients had adrenal disease compared to white and black patients.
- Table I – year of diagnosis – I wonder if it would be better to present the race distribution for each year rather than the distribution by year for each race. The distribution by year for each race is basically reflecting the years included in each category (1973-199 = many years, bigger cohort in each race, 2000-2004, few years, smallest cohort in each race). Since year of diagnosis is ultimately the significant finding it would more meaningful to know that the racial distribution was stable across the year cohorts.
- Discussion – line 176 – I would not say the results are “different”, I would say that they update and expand the results from previous SEER studies. The St Jude paper [11] is relatively old and therefore most of the improvements in survival captured in this manuscript were not captured in this older manuscript. The more recent reference [18] focuses on the role of SES on outcomes and highlights the mediating fore of SES when looking at race. I think that the first couple of sentences of the discussion could be modified a bit and acknowledge that the results are different because timeline or emphasis of the paper is different. I agree with the methodological differences described in discussion lines 200-210 but think there could be less emphasis at the beginning of the discussion in showing this manuscript as “different”.
- I think that the most important finding of this manuscript is the documentation of improved survival over time and the resolution of previously identified or suggested disparities by race. I think that the element of “narrowing of the previously observed racial disparity (line 203) could be emphasized in the abstract and the conclusions.
- Limitations should include that this paper is not taking SES into account, which we know can be a major driver of disparities.
Specific grammar or content suggestions to improve the manuscript are as follows:
- I would start the abstract by stating “Conclusive information….” Instead of “Information…” because we do have several papers on the subject matter, the problem is drawing conclusions from the available data (for all the reasons the authors explain it the into and discussion)
- Introduction – Line 56-59 – sentence could read: “Whereas some of the previous studies using SEER data revealed a reduced survival in Back patients [10,11], others did not coincide [9,12,14].” I realize the latter focus on mortality and not on survival, however, they are two faces of the same coin and the current working is complex to read through.
- Line 179 – duplicate word: “for patients’ black patients”
- Introduction – Line 56-59 – sentence could read: “…. Blacks had a 1.4-fold risk of death due to neuroblastoma, while Asians and Native Americans had a 1.6 and a 3-fold increased risk, respectively.”
- Lin 174-175 = I would remove the “yet”
- Line 224 – there is a “then” that should be a “than” at the end of the line
Author Response
Thank you very much for all your valuable comments and guidance to improve our manuscript. We have carefully studied your comments and revised our work according to your suggestions.
Comment#1: Insurance status – The element of insurance status is the only truly inconsistent element of the Methods/Results/Discussion. In the current version, insurance status appears as a finding in the discussion without provision of enough detail in the methods and results sections. I found it hard to determine if it should stay or be removed from the manuscript all together because the information provided is too limited and vague. If the authors want to retain this element, it should be mentioned as a variable in section 2.2 in greater detail. It seems that data is only available for 2007-2015, that should be specified there as well. Mentioning sub-analysis in section 2.3 needs a bit more detail, was this pursued only as univariate analysis? Was there a multivariate analysis of the 2007-2015 subset? Distribution should be shown in table 1 and results should be mentioned in the results section.
Response#1: We agree with the reviewer and have decided to take remove the description and discussion of insurance status from the manuscript to avoid confusion. We did not include it in the adjusted analysis as that information is only available from 2007 on-wards in SEER).
We have revised the manuscript accordingly.
Comment#2: Methods – SEER*Stat – Where the analyses done using SEER*STAT software or exported and analyzed in an external software. The software and version should be specified.
Response#2: We used Stata SE version 14.
Following sentence was added to the statistical methods section to provide this additional information:
“The data was exported from SEER and analyzed with Stata SE version 14.”
Comment#3: Results – The sentence that spans lines 126-128 on stage could be removed as those differences are not significant. I would highlight that a lower proportion of black patients were <1 year of age at presentation compared to White and API and that a higher proportion of API patients had adrenal disease compared to white and black patients.
Response#3: We have revised these lines accordingly. We have removed the description of the non-significant results and added the following sentences according to the suggestions by the reviewer:
“A lower proportion of black patients were <1 year of age at presentation compared with white and API. Also, a higher proportion of API patients had adrenal disease compared with white and black patients.”
Comment#4: Table I – year of diagnosis – I wonder if it would be better to present the race distribution for each year rather than the distribution by year for each race. The distribution by year for each race is basically reflecting the years included in each category (1973-199 = many years, bigger cohort in each race, 2000-2004, few years, smallest cohort in each race). Since year of diagnosis is ultimately the significant finding it would more meaningful to know that the racial distribution was stable across the year cohorts.
Response#4: Thank you very much for your comments. The reason why we decided to divide the time periods was in accordance with changes in the treatment protocols for neuroblastoma that occurred in the years 2000 and 2005 (https://www.ncbi.nlm.nih.gov/pmc/articles/PMC4567703/). To provide a better clarification, we have added the following information to the methods section where we describe the variable year of diagnosis:
“Pinto et al, showed that survival rates were determined to be significantly different according to diagnostic era with better outcome observed for patients diagnosed after 2000, when consolidation with HDT and stem-cell rescue was routinely included in the treatment plan for high-risk patients [16]. Only 6% (64 of 1,015) of the patients diagnosed between 2000 and 2004 and 30% (445 of 1,484) of those diagnosed between 2005 and 2010 received immunotherapy and cytokines plus isotretinoin after consolidation. Postconsolidation treatment with immunotherapy and cytokines plus isotretinoin is now considered part of standard-of-care treatment. Therefore, the categories of year of diagnosis, were the intervals of 1979 to 1999, 2000 to 2004, and 2005 to 2015 as to coincide with changes in treatment protocol for neuroblastoma implemented in 2000 and 2005 that could impact patient’s outcomes [16].”
Comment#5: Discussion – line 176 – I would not say the results are “different”, I would say that they update and expand the results from previous SEER studies. The St Jude paper [11] is relatively old and therefore most of the improvements in survival captured in this manuscript were not captured in this older manuscript. The more recent reference [18] focuses on the role of SES on outcomes and highlights the mediating fore of SES when looking at race. I think that the first couple of sentences of the discussion could be modified a bit and acknowledge that the results are different because timeline or emphasis of the paper is different. I agree with the methodological differences described in discussion lines 200-210 but think there could be less emphasis at the beginning of the discussion in showing this manuscript as “different”.
Response#5: The reviewer has a valid point. In agreement with your comment. Therefore, we have made several changes modifying the first couple of sentences. The first sentences read now as follows.
“Our results update and expand the results reported in previous studies using SEER [10,11,18]. A study analyzing survival between black and white patients with childhood cancer from 1992 to 2000 and from 2001 to 2007 revealed that despite significantly improved treatment outcomes, black patients had significantly poorer rates of survival in both study periods compared with white patients [11]. Moreover, Kehm et al found that black neuroblastoma patients exhibited 38 % higher risk of mortality compared with white patients after adjusting for age, sex and stage if diagnosis [18]. However, the emphasis of their paper was more on the role of socioeconomic status on outcomes and highlighted the mediating fore of socioeconomic status when looking at race.”
Comment#6: I think that the most important finding of this manuscript is the documentation of improved survival over time and the resolution of previously identified or suggested disparities by race. I think that the element of “narrowing of the previously observed racial disparity (line 203) could be emphasized in the abstract and the conclusions.
Response#6: We agree with the reviewer and have made the suggested changes.
We have revised the conclusion of the abstract and its states now as follows:
“No association between race and survival time was found. However, survival improved among all patients treated during 2000-2004 and 2005-2015 compared with those treated before the year 2000, leading to a narrowing of the racial disparity based on survival.”
The final three sentences in the conclusion section now read as follows:
“Finally, the change in standard of treatment after 2000 showed a statistically significant decrease in hazard ratio. Thus, survival improved among all patients treated during 2000-2004 and 2005-2015 compared with those treated before the year 2000, leading to a narrowing of the racial disparity based on survival. Improvements in treatment specific protocols should be pursued for all patients to benefit overall survival outcomes.”
Comment#7: Limitations should include that this paper is not taking SES into account, which we know can be a major driver of disparities.
Response#7: We agree with the reviewer. We have added following sentence to the limitation of the study section (second last paragraph of the discussion):
“Furthermore, we did not have access to information related to variables such as socioeconomic status and education. Social determinants may be the underlying factors in racial disparities of survival in neuroblastoma patients, and future research may study these factors in more detail.”
Comment#8: I would start the abstract by stating “Conclusive information….” Instead of “Information…” because we do have several papers on the subject matter, the problem is drawing conclusions from the available data (for all the reasons the authors explain it the into and discussion)
Response#8: We agree. We have revised the abstract and its starts now with “Conclusive information…”
Comment#9: Introduction – Line 56-59 – sentence could read: “Whereas some of the previous studies using SEER data revealed a reduced survival in Back patients [10,11], others did not coincide [9,12,14].” I realize the latter focus on mortality and not on survival, however, they are two faces of the same coin and the current working is complex to read through.
Response#9: We followed your advice and have revised the sentence. The new sentence reads now as suggested:
Whereas some of the previous studies using SEER data revealed a reduced survival in Back patients [10,11], others did not coincide [9,12,14]
Comment#10: Line 179 – duplicate word: “for patients’ black patients”
Response#10: Thanks, we have removed the duplicate word.
Comment#11: Introduction – Line 56-59 – sentence could read: “…. Blacks had a 1.4-fold risk of death due to neuroblastoma, while Asians and Native Americans had a 1.6 and a 3-fold increased risk, respectively.”
Response#11: Thank you. We have followed your advice and changed the sentence as suggested to”
“…blacks had a 1.4-fold risk of death due to neuroblastoma, while Asians and Native Americans had a 1.6 and a 3-fold increased risk, respectively.”
Comment#12: Lin 174-175 = I would remove the “yet”
Response#12: Thank you. We have removed the “yet”.
Comment#13: Line 224 – there is a “then” that should be a “than” at the end of the line
Response#13: Thank you. We have
Round 2
Reviewer 2 Report
Lines 94-95 of the manuscript say that "Patients who were alive at the date of the last contact or had a recurrence of neuroblastoma were censored."
This is indirect contradiction to the authors' response, which says, "As we did not censor patients at the time of recurrence of neuroblastoma, the curves are not misleading."
It is considered a fatal flaw to censor patients prior to the time when the patient could potentially die from neuroblastoma. It remains unclear what the authors actually did. The authors should remove the censoring of patients who had a recurrence of neuroblastoma. The only reason a patient should be censored is at the time of last contact when the patient has still been reported as alive. Patients with recurrence of neuroblastoma should continue to be followed, without censoring, until such time as the patient dies or has a date of last contact when the patient is alive. Until the analyses, or the statement of the methods, have been corrected, the results are not to be trusted.
It is unclear if a statistician has been involved in these analyses. If a statistician has not been involved, the authors are strongly encouraged to include a statistician in performing the analysis.
Author Response
Response#1: Dear reviewer, thank you very much for reading our manuscript in great detailed. We have corrected this statement now, so it agrees now with the lines 84-88 of our revised version. As stated previously, there was a misunderstanding due to how we described this in the manuscript. We meant that we excluded duplicate information but did not censor these patients. We discussed this with our statisticians we were working with.
In our previous revised version, we did correct this in lines 84-88, but we overlooked lines 94-95 and, thus, did not correct it accordingly. This has now been fixed, we apologize for that and are grateful to you to point this out.
Round 3
Reviewer 2 Report
Thank you for addressing the inconsistency in the description of the methods. I have no further comments.
This manuscript is a resubmission of an earlier submission. The following is a list of the peer review reports and author responses from that submission.
Round 1
Reviewer 1 Report
This study to investigate the influence of race on survival in neuroblastoma patients in the US addresses a common question in pediatric oncology and gives an important update by including more recent data than previous studies.
However, the reference list is extremely short and lacks important studies relevant to this topic, which would also help with ideas to present the results better, for example in display items (PMIDs 23243203, 31338261, 29117360 and many more).
In general, the SEER database is a solid basis for such analyses, but the presentation of the results makes it impossible to compare this study to previous analyses, which clearly show an association between race and survival in neuroblastoma patients in the US. Thus, the discussion section is very speculative and does not sound convincing.
Since the manuscript does not have many display items, it is not clear why the only figure presented shows the insurance status of neuroblastoma patients, when the study is about race and probably therapy differences. Kaplan Meier curves and corresponding tables addressing these parameters should be shown as, for example, in PMID23243203. With such visualization and time-dependent subgroup-specific data in tables, it would be possible to identify critical factors (such as small subgroup size, short follow up time or many others) that could have an effect on the detection of significant differences. Moreover, it would help to have hazard ratios displayed as forest plots.
Minor comments: Please check numbers (abstract, line 19, "n=2,119" vs MM, line 68, "2121"; MM, line 76, "15 to 17 years" should probably be "5 to 17 years").
Author Response
Thank you very much for all your valuable comments and guidance to improve our manuscript. We have carefully studied your comments and revised our work according to your suggestions. Your feedback helps us to become better researchers and are highly appreciated.
Comment#1: The reference list is extremely short and lacks important studies relevant to this topic, which would also help with ideas to present the results better, for example in display items (PMIDs 23243203, 31338261, 29117360 and many more).
Response#1: Thank you for the comments and valuable suggestions. We have identified additional studies in regard the association between race and survival in pediatric patients with neuroblastoma that have now been included in both the introduction and the discussion section. Some of the studies you mentioned such as Henderson et al. Pui et al and Johnson et al were already included in our first version of the manuscript. We have now addressed them better. Following new studies have been added:
- Li, X.; Meng, Y. A prognostic nomogram for neuroblastoma in children. PeerJ 2019, 11(7), e7316.
- Gamazon, E.R.; Pinto, N.; Konkashbaev, A.; Im, H.K.; Diskin, S.J.; London, W.B.; Maris, J.M.; Dolan, M.E.; Cox, N.J.; Cohn, S.L. Trans-population analysis of genetic mechanisms of ethnic disparities in neuroblastoma survival. J Natl Cancer Inst 2013, 105(4), 302-309.
- DiNicola, W.; Movassaghi, N.; Leikin, S. Prognosis in Black children with neuroblastoma. Cancer 1975, 36(3), 1151-1153.
- Friedrich, P.;Itriago E.; Rodriguez-Galindo, C.; Ribeiro, K.. Racial and Ethnic Disparities in the Incidence of Pediatric Extracranial Embryonal Tumors. J Natl Cancer Inst 2017, 109(10). doi: 10.1093/jnci/djx050.
We have also added a Forest Plot with a graphical display of our results of the adjusted Cox regression analysis file for those readers who prefer Figures over a Table. In addition, we have added the Kaplan-Meier survival curves according to race and added a risk-table below the Figure as done by the studies you have mentioned.
Comment#2: In general, the SEER database is a solid basis for such analyses, but the presentation of the results makes it impossible to compare this study to previous analyses, which clearly show an association between race and survival in neuroblastoma patients in the US. Thus, the discussion section is very speculative and does not sound convincing.
Response#2: We have revised the introduction and the discussion sections of the manuscript accordingly. Following changes were made in the introduction (changes are underlined):
Despite racial differences reported in the incidence of neuroblastoma, there is not much research on how race influences survival among children with neuroblastoma [8-14]. Results in regard differences in survival between white and black patients have been inconsistent [8, 11-14]. Whereas some of the previous studies using SEER data revealed a reduced survival in Black patients [10, 11], others did not reveal an increased risk of mortality in black compared with white children [9,12,14]. Results of A study of 3,539 children diagnosed with neuroblastoma and participating in the Children's Oncology Group neuroblastoma biology protocol between 2001 and 2009 showed that compared to whites, blacks had about 1.4-fold, while Asians and Native Americans had a 1.6 and a 3-fold increased risk of death due to neuroblastoma, respectively [8]. While blacks and whites were the most commonly included in studies, studies that also assess Native Americans & Asians [8,9], Hispanics [8,9,12], American Indians & Native Pacific Islanders [9,12] survival in neuroblastoma patients are scarce [8,9,10,12].
We re-wrote the second paragraph of the discussion and deleted the fourth paragraph in order to be consistent with the results of our study. We hope that the discussion is now easier to read and more convincing:
Our results are different than those reported in previous studies using SEER or other data [10,11,18]. A study analyzing survival between black and white patients with childhood cancer from 1992 to 2000 and from 2001 to 2007 revealed that despite significantly improved treatment outcomes for patients’ black patients had significantly poorer rates of survival in both study periods compared with white patients [11]. Moreover, Kehm et al found that black neuroblastoma patients exhibited 38 % higher risk of mortality compared with white patients after adjusting for age, sex and stage if diagnosis [18]. Finally, research comparing the survival rates by race among children with common extracranial solid tumors between 1985 and 2005 reported that black children had a 31% higher risk of death compared with White children [10]. Studies that used other data bases than SEER reported inconsistent results in regard an association between race and survival in children with neuroblastoma [8, 12, 13, 14]. In the early 1970thies, DiNicola et al compared the survival rates of 30 white and 15 black children with neuroblastoma treated at the Children’s Hospital National Medical Center in Washington DC [14]. They did not reveal a statistically significant difference between the white and black children regarding median duration of survival or percentage of long-term survivors [14]. Henderson et al showed a significantly increased risk of death from neuroblastoma in blacks compared to whites, before adjustment for risk groups [8]. After adjustment, overall survival did not differ between white and black patients. However, among patients who remained event-free for 2 years or longer, survival was worse in blacks compared with whites (HR 1.5, 95% CI 1.0-2.3) [8]. Also, a US cohort study including 2709 patients revealed that African genomic ancestry was associated with a 40% lower event-free survival demonstrating that common genetic variation influences neuroblastoma phenotype and contributes to the ethnic disparities in survival observed [13]. Finally, a recent study in patients diagnosed with neuroblastoma between 1986 and 2012 using data from the Therapeutically Applicable Research to Generate Effective Treatments initiative did not find any statistically significant differences in survival between black and white patients [12].
Comment#3: Since the manuscript does not have many display items, it is not clear why the only figure presented shows the insurance status of neuroblastoma patients, when the study is about race and probably therapy differences. Kaplan Meier curves and corresponding tables addressing these parameters should be shown as, for example, in PMID23243203. With such visualization and time-dependent subgroup-specific data in tables, it would be possible to identify critical factors (such as small subgroup size, short follow up time or many others) that could have an effect on the detection of significant differences.
Response#3: We agree with the reviewer. We have now replaced that previous Figure with the Kaplan Meier curves with a new Figure that shows the Kaplan Meier curves according to race. In agreement with the study PMID23243203, we have also added the risk table below the Figure that presents the number of patients at risk at 20, 40 and 60 months after diagnosis. Finally, according to your suggestion, we have now added a Forest plot presenting the findings of the adjusted survival analysis in Figure 2 for the most important variables. We wish to keep Table 2 as it outlines the numeric values of the strength of the association and the precision (95% confidence interval) for all variable in the unadjusted and adjusted analysis. We decided not to add age and tumor stage to Figure 2 as they point estimates are much higher than the rest and the Figure becomes to small as the scale needs to be adapted to higher hazard ratios. We hope this is acceptable for you.
Comment#4: Moreover, it would help to have hazard ratios displayed as forest plots.
Response#4: As outlined above, we have now added a Forest plot presenting the findings of the adjusted survival analysis in Figure 2 for the most important variables. We wish to keep Table 2 as it outlines the numeric values of the strength of the association and the precision (95% confidence interval) for all variable in the unadjusted and adjusted analysis. We decided not to add age and tumor stage to Figure 2 as they point estimates are much higher than the rest and the Figure becomes to small as the scale needs to be adapted to higher hazard ratios. We hope this is acceptable for you.
Comment#5: Minor comments: Please check numbers (abstract, line 19, "n=2,119" vs MM, line 68, "2121"; MM, line 76, "15 to 17 years" should probably be "5 to 17 years").
Response#5: Thank you. We have corrected the above-mentioned mistakes. The changes are marked in text in yellow.
Reviewer 2 Report
This paper describes the relationship between race and survival in children with neuroblastoma in the SEER database. The following issues have been found:
Lines 17-20, in the abstract, there were a couple of grammatical errors: "This was a retrospective…"; "The outcome variable was time of from diagnosis to death."
Line 51, reference style should be consistent throughout the manuscript, 7 (superscript) vs. [7].
Lines 19 and 68, there is a discrepancy in the reported sample size (n=2,119 in the Abstract vs. n=2,121 in the Methods).
Line 76, participant age category 4 to 15 years is missing, or there is a typo.
Lines 97-98, "The final adjusted model included age, gender, ethnicity, stage, site, and year of diagnosis." This is a result and should be presented in that section with the justification for the selection of these factors for inclusion in the final adjusted model.
Table 1, the header should clarify that the numbers in parentheses in the table (i.e., (724), (81), (72), etc.) represent counts: "% (n)". The "%" symbol is missing for the Black column.
Line 121, "… among for the study sample." Doesn't make sense; needs to be fixed.
Line 135, "among the" should be replaced with "by" or "between".
Figure 1 legend and text referring to the figure state that it represents adjusted survival curves by time period however the figure key refers to insurance status. This needs to be reconciled and the correct figures provided.
Lines 188-189, "… and is yet to be determined.," doesn't make sense; needs to be fixed.
A major limitation, since this was not a randomized study, is that there are other latent biological factors beyond those available in SEER that are known to affect survival in neuroblastoma patients and may be potentially confounded with race, such as MYCN, ploidy, histology, etc. There should be a bit more discussion about limitations in general with observational studies.
Author Response
Thank you very much for all your valuable comments and guidance to improve our manuscript. We have carefully studied your comments and revised our work according to your suggestions.
Comment#1: Lines 17-20, in the abstract, there were a couple of grammatical errors: "This was a retrospective…"; "The outcome variable was time of from diagnosis to death."
Response#1: Thank you. We have corrected the grammatical errors.
Comment#2: Line 51, reference style should be consistent throughout the manuscript, 7 (superscript) vs. [7].
Response#2: Thank you. We have corrected the reference style throughout the manuscript.
Comment#3: Lines 19 and 68, there is a discrepancy in the reported sample size (n=2,119 in the Abstract vs. n=2,121 in the Methods).
Response#3: Thank you. We have revised the numbers and correct them. They are now consistent throughout the manuscript (n=2119).
Comment#4: Line 76, participant age category 4 to 15 years is missing, or there is a typo.
Response#4: Thank you. We corrected the typo.
Comment#5: Lines 97-98, "The final adjusted model included age, gender, ethnicity, stage, site, and year of diagnosis." This is a result and should be presented in that section with the justification for the selection of these factors for inclusion in the final adjusted model.
Response#5: We agree with the reviewer and have moved that particular sentence to the result section where we present Table 2. We have also added a brief justification why these covariates were included in the adjusted model.
The final adjusted model included age, gender, ethnicity, stage, site, and year of diagnosis. These covariates were identified as being clinically relevant and were sued in previous studies as well [8-14]. In addition, they were unequally distributed according to the race (Table 1) and the survival.
Comment#6: Table 1, the header should clarify that the numbers in parentheses in the table (i.e., (724), (81), (72), etc.) represent counts: "% (n)". The "%" symbol is missing for the Black column.
Response#6: We agree. We have corrected Table 1 accordingly.
Comment#7: Line 135, "… among for the study sample." Doesn't make sense; needs to be fixed: "among the" should be replaced with "by" or "between".
Response#7: True. We have corrected that sentence.
Comment#8: Figure 1 legend and text referring to the figure state that it represents adjusted survival curves by time period however the figure key refers to insurance status. This needs to be reconciled and the correct figures provided.
Response#8: We have revised the legend accordingly. It states now as follows:
Figure 1. Adjusted survival curves according to race for patients in the SEER database diagnosed with neuroblastoma, 1973-2015.
Comment#9: Lines 188-189, "… and is yet to be determined.," doesn't make sense; needs to be fixed.
Response#9: We have revised that sentence. It reads now as follows:
Possible reasons for the varying results could have been due to the different categorization and use of patient characteristics, neuroblastoma sites, time periods and the use of different databases.
Comment#10: A major limitation, since this was not a randomized study, is that there are other latent biological factors beyond those available in SEER that are known to affect survival in neuroblastoma patients and may be potentially confounded with race, such as MYCN, ploidy, histology, etc. There should be a bit more discussion about limitations in general with observational studies.
Response#10: We agree with the reviewer and have added more information to the limitation section. We have added the following lines to the limitations of the study section to address the concerns in regard our research design.
Lastly, there are some latent biological factors beyond those available in SEER that are known to affect survival in neuroblastoma patients and may be potentially confounded with race, such as MYCN, ploidy and tumor histology, among others. Randomized clinical trials are better study designs to control for confounders than observational studies. However, due to the nature of our exposure a randomized clinical trial is not applicable for our research question. Thus, best scientific evidence needs to be derived from future cohort studies that may control for most of the above-mentioned confounders we did not have available.